# Surpassing Cosine Similarity for Multidimensional Comparisons: Dimension Insensitive Euclidean Metric

## Abstract

Advances in computational power and hardware efficiency have enabled tackling increasingly complex, high-dimensional problems. While artificial intelligence (AI) achieves remarkable results, the interpretability of high-dimensional solutions remains challenging. A critical issue is the comparison of multidimensional quantities, essential in techniques like Principal Component Analysis. Metrics such as cosine similarity are often used, for example, in the development of natural language processing algorithms or recommender systems. However, the interpretability of such metrics diminishes as dimensions increase. This paper analyzes the effects of dimensionality, revealing significant limitations of cosine similarity, particularly its dependency on the dimension of vectors, leading to biased and poorly interpretable outcomes. To address this, we introduce a Dimension Insensitive Euclidean Metric (DIEM), which demonstrates superior robustness and generalizability across dimensions. DIEM maintains consistent variability and eliminates the biases observed in traditional metrics, making it a reliable tool for high-dimensional comparisons. An example of the advantages of DIEM over cosine similarity is reported for a large language model application. This novel metric has the potential to replace cosine similarity, providing a more accurate and insightful method to analyze multidimensional data in fields ranging from neuromotor control to machine learning.

## 1 Introduction

The growth of computational capability and improvement of hardware efficiency (Chen, 2016) enabled approaching problems of growing complexity and dimensionality. Over a little more than two decades, Artificial Intelligence (AI) agents defeated humans in games previously considered dominated by humans: chess (Hsu, 2022), Go (Silver et al., 2017), StarCraft II (Vincent, 2019). This ability to handle complex and high-dimensional problems also led to promising results in fields such as molecular biology (Jumper et al., 2021) or robotics (Kober et al., 2013). On one hand, the results of these computing techniques are undeniably impressive; on the other hand, their interpretability decreases with their complexity and dimensionality. One prominent concern is the comparison of multidimensional quantities. For example, in dimensionality reduction techniques, such as Principal Component Analysis (PCA), Singular Value Decomposition (SVD), or k-means clustering, the algorithms extract significant combinations of a selected set of input features. The number of input features can vary greatly, from tens – e.g., the joint angles of the hand (West et al., 2023) – to thousands as with text embeddings of Natural Language Processing (NLP) algorithms, and Large Language Models (LLMs) (OpenAI, 2025). A reasonable question is: "How similar or different are these combinations of features?"

Consider, for instance, the task of identifying an individual user's preference for movies based on input features such as the movies the user has watched, his/her evaluation of them, and the date when they were watched (Bennett & Lanning, 2007). Alternatively, consider the task of identifying during e.g., grasping an object, how an individual coordinates his/her joints (shoulder, elbow, wrist, phalanges) to successfully complete the task (Santello et al., 1998; Mason et al., 2001). Assuming the algorithm identifies a specific individual's preferences, how can we compare two different individuals' preferences? As long as the number of input features is limited to 2 or 3, humans are visually

able to interpret them using a planar or spatial analogy. However, when feature spaces go beyond 3D, visualization and interpretation become more complex and less intuitive. As a consequence, researchers typically rely on mathematical measures to gain an understanding of the similarity (or difference) between n-dimensional quantities. Among the many methods for multidimensional comparisons, cosine similarity stands out as one of the 'gold standards' (Ye, 2011; Lahitani et al., 2016; Xia et al., 2015; Nguyen & Bai, 2011; Luo et al., 2018; Eghbali & Tahvildari, 2019). This is likely related to its analogy with angular measurements and its bounded, well-defined range (between 0 and 1, or -1 and 1, depending on formulation).

In this work, we present a detailed analysis of the effects of dimensionality on cosine similarity. Some interesting limitations and properties emerge, leading to the conclusion that the use of cosine similarity might not be the most appropriate choice for multidimensional comparisons. An alternative – named DIEM (Dimension Insensitive Euclidean Metric) – is proposed, which shows better robustness and generalizability to increasing numbers of dimensions. The advantages of DIEM over cosine similarity are showcased by a comparison of text embeddings from an existing open-source large language model. The new metric proves to surpass cosine similarity for high-dimensional comparisons.

## 2 COSINE SIMILARITY AND EUCLIDEAN DISTANCE

Given two vectors $\mathbf{a} = [a_1, \ldots, a_n]$ and $\mathbf{b} = [b_1, \ldots, b_n]$, with $a_i, b_i \in \mathbb{R}$, the cosine similarity between these vectors is defined as:

$$\cos(\theta) = \frac{|\mathbf{a}^T \cdot \mathbf{b}|}{\|\mathbf{a}\| \cdot \|\mathbf{b}\|} \tag{1}$$

This value ranges between 0 and 1, with 0 meaning the two vectors are orthogonal, and 1 meaning the vectors are collinear. We can consider the vectors $\mathbf{a}$ and $\mathbf{b}$ as points in an $n$-dimensional space and compute the Euclidean distance (2-norm) between them as:

$$d = \sqrt{\sum_{i=1}^{n}(a_i - b_i)^2} \implies d^2 = \sum_{i=1}^{n} a_i^2 + \sum_{i=1}^{n} b_i^2 - 2\sum_{i=1}^{n} a_i b_i \tag{2}$$

We can then reformulate Equation 1 in index notation:

$$\cos(\theta) = \frac{|\mathbf{a}^T \cdot \mathbf{b}|}{\|\mathbf{a}\| \cdot \|\mathbf{b}\|} = \frac{|\sum_{i=1}^{n} a_i b_i|}{\|\mathbf{a}\| \cdot \|\mathbf{b}\|} \tag{3}$$

Remembering that the Euclidean norm of a vector is equal to $\|\mathbf{x}\| = \sqrt{\sum_{i=1}^{n} x_i^2}$, and combining Equation 3 and Equation 2, we obtain:

$$\cos(\theta) = \frac{1}{\|\mathbf{a}\| \cdot \|\mathbf{b}\|} \cdot \left| \frac{\|\mathbf{a}\|^2 + \|\mathbf{b}\|^2 - d^2}{2} \right| \tag{4}$$

Equation 4 shows that the cosine similarity has a quadratic correlation with the Euclidean distance. Moreover, in the case in which our vectors $\mathbf{a}, \mathbf{b}$ have unit length, this relation simplifies to:

$$\cos(\theta) = \left| 1 - \frac{d^2}{2} \right| \tag{5}$$

For unit-length vectors, the Euclidean distance can only range between $0 \leq d \leq 2$, since the vectors—independently of the dimension of space that they span—can at most be on a diameter of a hypersphere of radius 1.

An alternative definition of cosine similarity excludes the absolute value of the scalar product in Equation 1. This results in the cosine similarity ranging from $-1$ to 1, thereby accounting for sign differences in the vectors. The relation with the Euclidean distance is presented in an Appendix.

## 3 EFFECT OF VECTOR DIMENSIONALITY

An interesting aspect is the sensitivity of the cosine similarity metric to the dimension ($n$) of the considered vectors as well as to the space spanned by them: $\mathbb{R}^n, \mathbb{R}^{n+}, \mathbb{R}^{n-}$. To test the sensitivity to these two aspects, a numerical simulation was run using the algorithm presented in Figure 1. All the numerical simulations were performed using Matlab 2023b on a laptop computer with 16 GB of RAM and an 11th Gen. Intel i7-11800H.

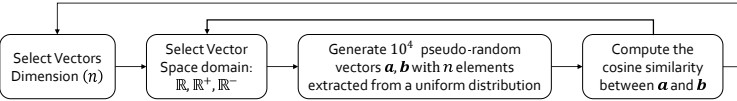

Figure 1: Algorithm used for the sensitivity analysis of cosine similarity with respect to vector dimension and domain.

The idea was to iterate through growing dimensions while comparing two randomly generated vectors ($\mathbf{a}, \mathbf{b}$). The vectors were generated from a pseudo-uniform distribution in order to avoid bias in the mean value of each vector's elements. The cosine similarity was computed for a range of dimensions going from 2 (planar case) to 102. Vectors $\mathbf{a}, \mathbf{b}$ were scaled such that, in the three different domains, their lengths assumed the following range of values $\mathbb{R} \rightarrow -1 \leq a_i, b_i \leq 1, \quad \forall i, \mathbb{R}^+ \rightarrow 0 \leq a_i, b_i \leq 1, \quad \forall i, \mathbb{R}^- \rightarrow -1 \leq a_i, b_i \leq 0, \quad \forall i$. These ranges were arbitrarily chosen to resemble the activation level of some feature, e.g., a surface electromyographic signal scaled to the maximum voluntary activation, or the text embedding vector of a large language model. A sensitivity analysis showed that vector scaling produced no effect on cosine similarity. The resulting cosine similarities for each vector dimension and each vector domain are presented in Figure 2.a.

Interestingly, the growing dimensionality of these random vectors led to a convergence of the average cosine similarity. For the only-positive or only-negative vectors, cosine similarity rapidly converged to a value of about $0.75 \pm 0.1$. For the vectors that could span both negative and positive values, cosine similarity converged to a value approaching 0. The Appendix presents a theoretical proof of the convergence of cosine similarity in all cases.

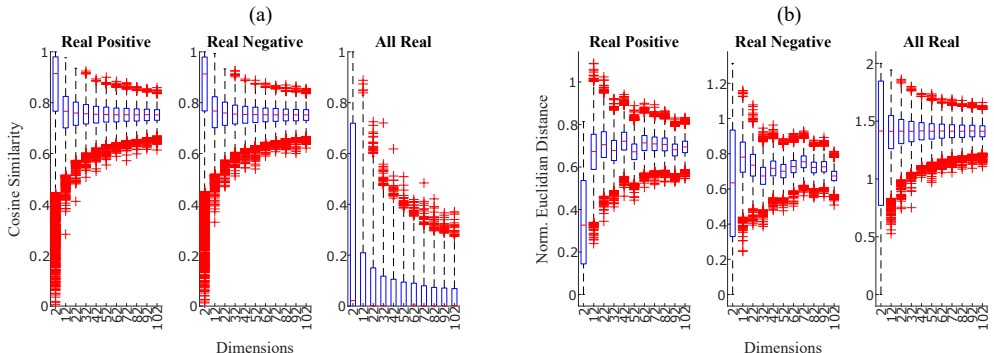

Figure 2: Panel (a): Cosine similarity boxplots for increasing dimension of the vectors $\mathbf{a}$ and $\mathbf{b}$. Panel (b): Normalized Euclidean distance boxplots for increasing dimension of the vectors $\mathbf{a}$ and $\mathbf{b}$. The three sub-panels show, respectively, the case in which vector elements were only positive (left), only negative (center) or could assume all real values within the given range (right).

The numerical analysis was repeated using the Euclidean distance between normalized vectors (Figure 2.b). Again, the normalized Euclidean distance converged toward a constant value with reduced variability. Specifically, for the real positive and negative cases, the metric converged to about $0.7 \pm 0.1$, while for the all-real case, it converged to $1.41 \pm 0.2$. This value is close to $\sqrt{2}$, the expected distance between two unit vectors that are perfectly orthogonal to each other.

A more troubling aspect, emerging from both the cosine similarity and normalized Euclidean distance (Figure 2), was the fact that the variability of these metrics was a strong function of the number of dimensions ($n$). Specifically, the variability of these random comparisons tended to narrow with an

increasing number of dimensions. Different distributions for sampling the vectors $\mathbf{a}$, $\mathbf{b}$, (Gaussian or uniform spherical) were tested, showing equivalent results to those presented in Figure 2; see Appendix. If, instead, we consider the non-normalized Euclidean distance, we observe the behavior presented in Figure 3.a.

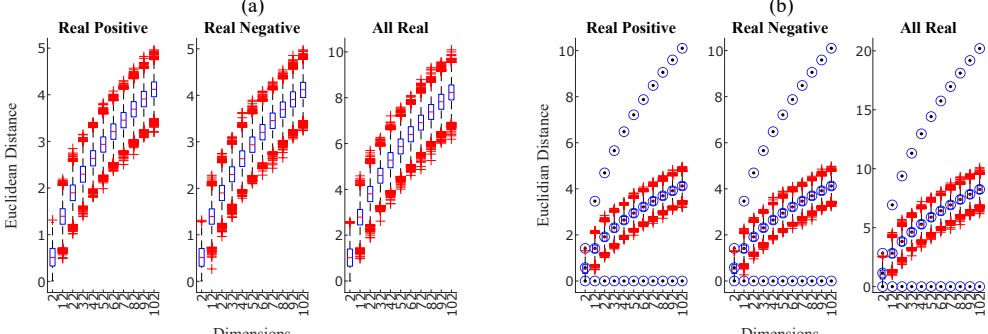

Figure 3: Panel (a): Euclidean distance for increasing dimension of the vectors $\mathbf{a}$ and $\mathbf{b}$. The three sub-panels show, respectively, the case in which vectors elements were only positive (left), only negative (center) or could assume all real values within the given range (right). Panel (b): Euclidean distance for increasing dimension of the vectors $\mathbf{a}$ and $\mathbf{b}$. The blue circles show the minimum, maximum and expected analytical Euclidean distance values.

The center of the Euclidean distance distribution tended to grow without bound, but the variability of each boxplot remained approximately constant. This is an improvement over the cosine similarity or normalized Euclidean distance case for two reasons: (i) the variability does not change with the dimension of the vectors $\mathbf{a}$ and $\mathbf{b}$, i.e., the variance ($\sigma_{ed}^2$) is only a function of the range $v_m, v_M$ but not of the number of dimensions $n$, and (ii) the metric does not converge towards a plateau.

## 4 MATHEMATICAL PROPERTIES OF THE EUCLIDEAN DISTANCE

Some interesting properties can be analytically derived from the definition of Euclidean distance (Equation 2), assuming that each element of any two vectors $\mathbf{a}$, $\mathbf{b}$ is bounded between a minimum $v_m$ and maximum value $v_M$, with $v_M > v_m$. Boundedness is a reasonable assumption, since measurable physical quantities are typically bounded either by physical or measurement limitations. Considering these assumptions, the minimum Euclidean distance between any two $n$-dimensional vectors $\mathbf{a}$ and $\mathbf{b}$ is always zero, independent of dimension:

$$d_{\min}(n) = 0 \quad \forall n \tag{6}$$

This is trivially demonstrated by assuming that: $\forall i = 1, \ldots, n \quad \Rightarrow \quad a_i = b_i$.

The maximum Euclidean distance can be computed considering that—for a given dimension $n$—each element of the vector $\mathbf{a}$ assumes its maximum value $v_M$, while each element of the other vector $\mathbf{b}$ assumes its minimum value $v_m$. At that point, the Euclidean distance from Equation 2 becomes:

$$d = \sqrt{\sum_{i=1}^{n}(a_i - b_i)^2} \leq \sqrt{\sum_{i=1}^{n}(v_M - v_m)^2} = \sqrt{n(v_M - v_m)^2} \Rightarrow d_{\max}(n) = \sqrt{n}\cdot(v_M - v_m) \tag{7}$$

Moreover, assuming the elements of the two vectors ($\mathbf{a}$, $\mathbf{b}$) are sampled from two independent and identically distributed (i.i.d.) random variables with uniform distributions $a_i \sim U(v_m, v_M), \quad b_i \sim U(v_m, v_M)$, the upper bound of the expected Euclidean distance can be derived analytically, i.e., the

main trend observed in Figure 3.a. Specifically, by applying Jensen's inequality[1] (Jensen, 1906) and the "Law Of The Unconscious Statistician (LOTUS)" (Groot & Schervish, 2014), we obtain:

$$E[d] = E\left[\sqrt{\sum_{i=1}^{n}(a_i - b_i)^2}\right] \leq \sqrt{E\left[\sum_{i=1}^{n}(a_i - b_i)^2\right]}$$

$$= \sqrt{n \cdot E[(a-b)^2]} = \sqrt{n}\left(\iint_{v_m}^{v_M} \frac{(a-b)^2}{(v_M - v_m)^2}\, da\, db\right)^{\frac{1}{2}} \tag{8}$$

The integral in Equation 8 can easily be solved by direct integration. We leave readers the pleasure to do so. The final result can be expressed in the following form:

$$E[d] \leq \sqrt{n}\sqrt{\frac{2}{3}(v_M^2 + v_M v_m + v_m^2) - \frac{1}{2}(v_M + v_m)^2} = \sqrt{\frac{n}{6}}(v_M - v_m) \tag{9}$$

Equation 9 provides an analytical upper bound to the expected Euclidean distance between any two random vectors $\mathbf{a}, \mathbf{b} \sim U(v_m, v_M)$. Figure 3.b provides a graphical representation of the Euclidean distance along with its maximum (Equation 7), minimum (Equation 6), and expected (Equation 9) value across dimensions.

Interestingly, the analytical expected value of the Euclidean distance (Equation 9) is practically indistinguishable from the numerically simulated Euclidean distance median value for any dimension $n > 2$. Moreover, it is also interesting to observe that the expected value is smaller than the arithmetic mean between the maximum and minimum Euclidean distances:

$$\frac{d_{\min}(n) + d_{\max}(n)}{2} = \frac{\sqrt{n}}{2}(v_M - v_m) \Rightarrow \frac{\sqrt{n}}{2}(v_M - v_m) > \sqrt{\frac{n}{6}}(v_M - v_m) \tag{10}$$

Another interesting aspect to observe is the evolution of the Euclidean distance distribution for growing dimensions (Figure 4.a). The Real Positive case was considered, but equivalent results were obtained for the Real Negative and All Real cases. For each of the simulated dimensions ($n$), a Kolmogorov-Smirnov test was performed to check whether the observed simulated distribution was significantly different from a normal distribution with mean and standard deviation equal to those of the simulated data, i.e., $\mathcal{N}(\text{mean}(d), \text{std}(d))$. Interestingly, the test showed a significant difference only for $n = 2$ ($p < 0.05$); while, for all the other tested dimensions, the distribution of Euclidean distances was not significantly different from a normal distribution.

A finer simulation on a subset of dimensions ($2 \leq n \leq 12$) revealed a smooth transition between non-normal and normal distribution around $n \approx 7$; see Appendix.

A confirmation of these results can be found in the work of Thirey and Hickam (Thirey & Hickman, 2015), who analytically derived the distribution of Euclidean distances between randomly distributed Gaussian vectors and observed a normalization of the distribution with an increasing number of dimensions $n$.

Despite this trend toward normality in the distribution of Euclidean distances, it is important to emphasize that the distribution tails are not symmetric. This is clearly observable from Figure 3.b, and it is also demonstrated by Equation 10, since the expected value $E[d]$ is smaller than the mean between the maximum and minimum Euclidean distances. However, though the distribution is not strictly normal, it is indistinguishable from normal.

---

[1]Jensen's Inequality assumes concavity, and the square root function is indeed concave. This accounts for the 'less-than-or-equal' in Equation 8.

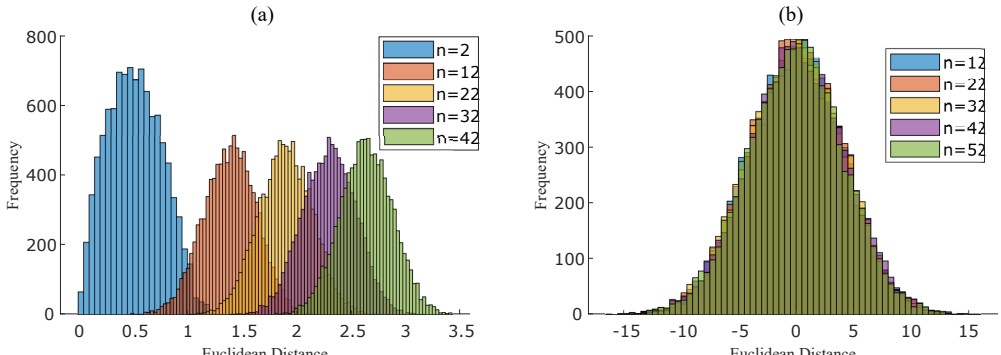

Figure 4: Panel (a): Histograms of the non-normalized Euclidean distance for growing dimensions 'n'. Panel (b): Histograms of the detrended Euclidean distance for growing dimensions 'n'.

## 5 A DIMENSION-INSENSITIVE METRIC FOR MULTIDIMENSIONAL COMPARISON

At this point, we can obtain a dimension-insensitive metric by subtracting the expected value $E[d(n)]$ from each Euclidean distance distribution, normalizing it by the variance $\sigma_{ed}^2$ of the Euclidean distance $\sqrt{\sum_{i=1}^{n}(a_i - b_i)^2}$, and scaling it to the range of the analyzed quantities $(v_M - v_m)$. This metric is defined as the Dimension Insensitive Euclidean Metric (DIEM):

$$\text{DIEM} = \frac{v_M - v_m}{\sigma_{ed}^2} \left( \sqrt{\sum_{i=1}^{n}(a_i - b_i)^2} - E[d(n)] \right) \tag{11}$$

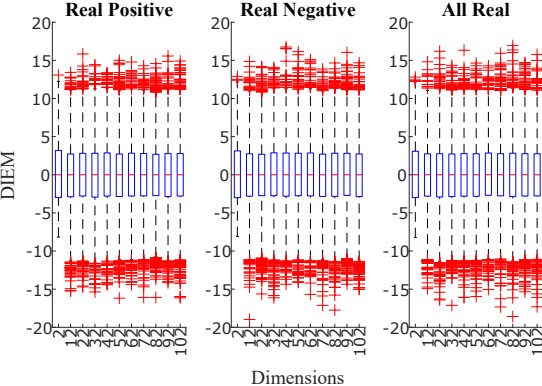

Figure 5: Dimension Insensitive Euclidean Metric (DIEM) for increasing dimension of the vectors **a** and **b**. The three panels show, respectively, the case in which vector elements were only positive (left), only negative (center) or could assume all real values within the given range (right).

The distributions of DIEM are presented in Figure 5, which shows that increasing the number of dimensions does not affect either the converged value or the variance of the measures, thus providing a more reliable comparison metric for high-dimensional vectors. Moreover, the histograms of the Dimension Insensitive Euclidean Metric demonstrate that the distributions are practically identical for $n \geq 7$ (Figure 4.b). This facilitates precise statistical testing.

A more intuitive understanding of similarity and dissimilarity using DIEM is provided in Figure 6. In general, vectors may have different magnitudes and orientations. Lower values of DIEM represent similar vectors, while higher values of the metric represent dissimilar vectors. Since

DIEM is detrended, its expected value $E[\text{DIEM}]$ is 0. The shaded red areas respectively show one, two, and three standard deviations ($\sigma$). Three dotted lines are added to indicate: (i) the maximum Dimension Insensitive Euclidean Metric ($\text{DIEM}_{\max}$), which represents antiparallel vectors with maximum magnitude, (ii) the expected Dimension Insensitive Euclidean Metric for orthogonal vectors, $E[\text{DIEM}_{\text{orth}}]$, and (iii) the minimum Dimension Insensitive Euclidean Metric ($\text{DIEM}_{\min}$), which represents identical vectors.

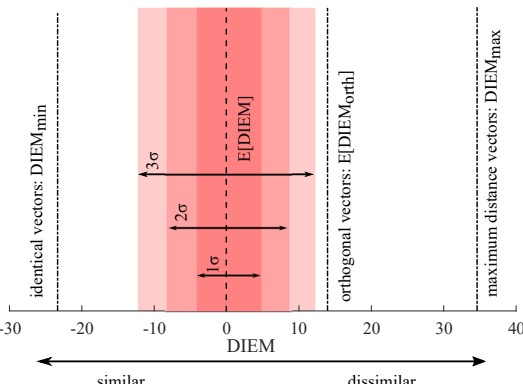

Figure 6: Explanation of the Dimension Insensitive Euclidean Metric (DIEM). This image was generated considering $n = 12$, $v_M = 1$, $v_m = 0$, $\sigma_{ed}^2 = 0.06$. The dashed black line represents the DIEM expected for random vectors. The red-colored bands represent respectively one ($\sigma = 4.09$), two, and three standard deviations of the DIEM distribution. The right-most dotted black line is the maximum $DIEM_{max} = 34.61$, representing two opposing vectors with maximum magnitude. The second from right dotted black line is the median value of DIEM for orthogonal vectors $E[DIEM_{orth}] = 13.95$. The left-most dotted black line is the minimum $DIEM_{min} = -23.35$, representing two identical vectors.

The mean Dimension Insensitive Euclidean Metric between orthogonal vectors ($E[\text{DIEM}_{\text{orth}}]$) was computed by generating a series of orthogonal vectors and numerically computing their expected value. From the considered case in Figure 6 ($n = 12$, $v_M = 1$, $v_m = 0$, $\sigma = 4.09$), the probability of two random vectors being orthogonal was more than three standard deviations from the mean.

The minimum DIEM is easily obtained by combining Equation 6 with Equation 11:

$$\text{DIEM}_{\min} = -\frac{v_M - v_m}{\sigma_{ed}^2} E[d(n)] \tag{12}$$

while the maximum DIEM is obtained by considering Equation 7 in combination with Equation 11:

$$\text{DIEM}_{\max} = \frac{v_M - v_m}{\sigma_{ed}^2} \left( \sqrt{n} \cdot (v_M - v_m) - E[d(n)] \right) \tag{13}$$

It is worth emphasizing that, while the expected value of the Dimension Insensitive Euclidean Metric ($E[\text{DIEM}]$) as well as its variance (and standard deviation) remain constant independent of the number of dimensions, the values of the maximum and minimum Dimension Insensitive Euclidean Metric ($\text{DIEM}_{\max}, \text{DIEM}_{\min}$), as well as the expected distance between orthogonal vectors ($E[\text{DIEM}_{\text{orth}}]$), are functions of $E[d(n)]$ and, thus, of the number of dimensions. Both quantities are, however, numerically computable either by numerical simulation ($E[\text{DIEM}_{\text{orth}}]$) or by direct derivation ($\text{DIEM}_{\max}, \text{DIEM}_{\min}$). Code is available at `https://anonymous.4open.science/r/ DIEM-5D68/README.md`

# 6 A CASE STUDY: LLMS TEXT EMBEDDINGS

Cosine similarity is widely used in Large Language Models (LLMs). Here we compare it with DIEM to showcase the advantages of DIEM over cosine similarity. A critical element of Large

Language Models (LLMs) are their text embedding models (OpenAI, 2025). Their function is to convert portions of the text (words, sentences, or entire documents) into numerical representations — typically vectors — that capture the semantic meaning and relationships within the text input.

Several embedding models exist and can be classified into three main categories: word-level (Word2Vec,GloVe), sentence-level (all-MiniLM-L6-v2 (Wang et al., 2020)) and document-level (BERT, GPT) embedders. Each model transforms text input (of variable length) into a fixed-length numerical vector. Word-level embedders typically output vectors of dimension $n = 300$, sentence-level embedders have $n = 384$, and document-level embedders have up to $n = 3072$ for large GPT models (OpenAI, 2025).

Text embedding models are necessary to perform most of the natural language processing (NLP) tasks, such as text generation or information retrieval. It is important to evaluate the similarity (or distance) between the output vector embeddings, thus providing us with a practical use-case to show the difference between cosine similarity and DIEM.

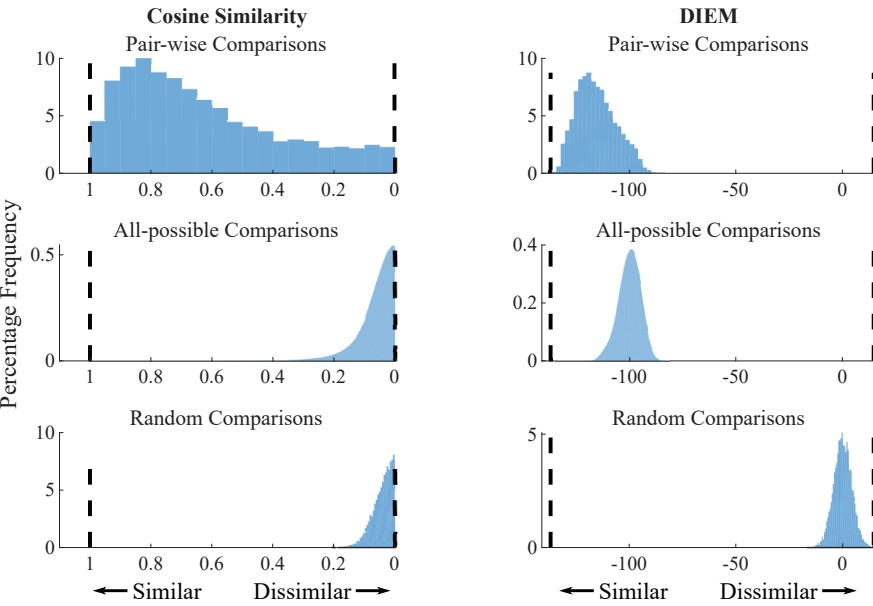

Figure 7: Histograms representing the similarity between text embeddings using cosine similarity (left column) and DIEM (right column), considering "pair-wise comparisons" (first row), "all-possible comparisons", and "random comparisons". The histograms are normalized to the total number of samples. The vertical black dashed lines show the most similar (left) and most dissimilar (right) sentences (vector embeddings) in the whole dataset. The vectors embeddings could range over all real values, but the cosine similarity was constrained between 0 and 1 as per Equation 1.

We used an existing language model, 'all-MiniLM-L6-v2' developed by Wang et al. (2020) to generate numerical vector embeddings ($n = 384$) from an existing dataset of sentences, i.e., the "Semantic Textual Similarity Benchmark" developed by Cer et al. (2017) and available at: `https://huggingface.co/datasets/sentence-transformers/stsb#dataset-details`. This dataset provides a fairly large ($8,628$ entries) collection of sentence pairs drawn from news headlines, video and image captions, and natural language inference data. A pair could be: "Three men are playing chess" - "Two men are playing chess." We computed both Cosine Similarity and DIEM between the generated vector embeddings for each pair ($8,628$ comparisons) and between all possible entries ($74,442,384$ comparisons). Moreover, we also generated an additional dataset of $10^4$ random vectors of dimension $n = 384$ spanning the same range ($v_{min}, v_{max}$) of the text embeddings. Note that the vector embeddings were not normalized to avoid the issues related to vector normalization highlighted in Figure 2.

Figure 7 shows histograms of the similarity between vector embeddings using Cosine Similarity and DIEM for 'pair-wise comparisons','all-possible comparisons', and 'random comparisons'. The Cosine Similarity converges from a modal value of about 0.8 in the 'pair-wise comparisons' case to a mode at 0 in the 'all-possible comparisons' case. Then, it maintains the same mode in the 'random comparisons' case. This represents a strong limitation of Cosine Similarity as a metric for comparison of large dimension ($n = 384$ in this case) big datasets ( 74 million points). On the other hand, DIEM does not present this drastic change when moving from the 'pair-wise comparisons' to the 'all-possible comparisons'. Instead, we observe a smooth distribution well clear of the ends of its range with its mode shifted from about -120 to about -100. This indicates a larger dissimilarity in the 'all-possible comparisons' case compared to the 'pair-wise comparisons' case, as would be expected. However, when moving from the 'all-possible comparisons' to the 'random comparisons', we see a further difference - not present in the Cosine Similarity case. Interestingly, a z-test between the 'all-possible comparisons' case and the known 'random comparisons' case reveals a significant difference between the two ($p \ll 0.001$). This suggests that the data in the 'all-comparisons case' are indeed less similar but still statistically distinguishable from random comparisons. The same could not be said for cosine similarity. In the 'all-possible comparisons' case, cosine similarity is statistically indistinguishable from 0, its expected value for purely random vectors in high dimensions.

## 7 DISCUSSION

In the study of multidimensional quantities (e.g., principal components extracted through PCA), cosine similarity is widely used as a metric to assess similarity between such quantities. Our results show that this metric is strongly influenced by the dimension of the vectors. This means that randomly generated vectors in high dimensions will result in a constant orientation with respect to each other. This is a completely misleading way to compare quantities, as it risks leading a naïve investigator to consider two vectors to be very different (or very similar) based on pure chance.

Similar considerations apply to many different fields, including recommender systems, data mining algorithms, and computational neuroscience models, in which high-dimensional quantities are commonly generated. This emphasizes the importance of a dimension-independent measure to correctly assess whether two vectors are similar (collinear) or dissimilar (orthogonal). By detrending the Euclidean distance boxplots (Figure 5) using the expected value we obtain a metric to compare high-dimensional vectors that is independent of the number of dimensions ($n$). This feature appears to be a particular property of the Euclidean distance. Using a different norm (Manhattan distance - see Appendix), the resulting behavior does not guarantee a dimension-insensitive variance (Figure 6). Moreover, for $n \geq 7$, for vectors selected at random, the distribution of detrended Euclidean distances is statistically indistinguishable from a normal distribution (Figure 4.b). This enables access to a broad set of rigorous statistical tests for comparisons, e.g., $t$-tests or ANOVA.

The analysis and comparison of multidimensional quantities—such as synergies in the human neuromotor control literature, principal components and clusters in deep learning methods, or vector embeddings in large language models—have long been a thorny and often neglected problem. Most studies rely on comparison metrics, such as cosine similarity, without a complete understanding of how they depend on the number of dimensions of the considered features. Our study revealed that this dimensional dependency can significantly bias the interpretation of these metrics, leading to potentially erroneous conclusions.

In response to this challenge, we introduce the Dimension Insensitive Euclidean Metric (DIEM), which effectively mitigates the dimensional dependency problem, providing a more accurate and interpretable measure for multidimensional comparisons. As may be expected, this metric does have limitations: it is unsuitable for multidimensional comparisons of normalized (unit-length) quantities (see Figure 2.b). If the problem context calls for normalization, this metric is inappropriate. Nonetheless, by avoiding normalization and adopting DIEM, researchers can enhance the reliability of their multidimensional analyses, paving the way for more precise and meaningful interpretations in fields including language processing, data mining, recommender systems, and computational neuroscience. Future research should continue to explore and refine this metric, ensuring its broad applicability and further validating its advantages over traditional methods.

## REPRODUCIBILITY STATEMENT

We are committed to ensuring the reproducibility of our results and have taken the following steps to support this goal:

- **Code Availability:** We provide the full source code for computing the DIEM metric, generating all experiments, and reproducing the figures and tables in the paper. The code is structured, documented, and is publicly available at: `https://anonymous.4open.science/r/DIEM-5D68/README.md`.

- **Data:** All experiments are based on publicly available datasets. Instructions for downloading and preprocessing these datasets are provided. When synthetic data is used, we include the code to generate them.

- **Experimental Setup:** All hyperparameters, evaluation protocols, and random seeds are specified in the paper or configuration files. Scripts to launch all experiments are provided.

- **Statistical Reporting:** For all experiments, we report averages over multiple runs and include standard deviations where appropriate. Sampling procedures and seed settings are clearly documented.

We believe these efforts ensure that our results are fully reproducible and that the DIEM metric can be evaluated and extended by the community.

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

# A    SUPPLEMENTARY MATERIAL

## A.1    COSINE SIMILARITY AND EUCLIDEAN DISTANCE

Figure 8 provides a graphical representation of the relation between cosine similarity and Euclidean distance for unit length vectors as well as a geometrical interpretation of their relationship.

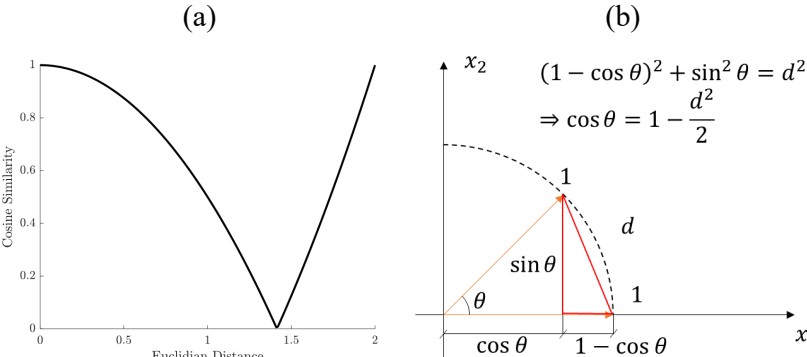

Figure 8: Panel (a): Graphical representation of the relation between cosine similarity and Euclidean distance for unit length vectors. Panel (b): Geometrical relation between the angle spanned by two unit vectors and the Euclidean distance between their end points.

## A.2 SIGNED COSINE SIMILARITY

The cosine similarity can also be expressed as:

$$\cos(\theta) = \frac{\mathbf{a}^T \cdot \mathbf{b}}{\|\mathbf{a}\| \cdot \|\mathbf{b}\|} \tag{14}$$

where the vectors $\mathbf{a}$ and $\mathbf{b}$ are defined as: $\mathbf{a} = [a_1, \ldots, a_n], \quad \mathbf{b} = [b_1, \ldots, b_n], \quad$ with $a_i, b_i \in \mathbb{R}$.

In this case, the cosine similarity spans from $-1$ (opposing vectors) to $1$ (aligned vectors), with $0$ representing orthogonal vectors. The mathematical relationship between this alternative definition of cosine similarity and the Euclidean distance remains equivalent. Equations 4 and 5 become:

$$\cos(\theta) = \frac{1}{\|\mathbf{a}\| \cdot \|\mathbf{b}\|} \cdot \frac{\|\mathbf{a}\|^2 + \|\mathbf{b}\|^2 - d^2}{2} \tag{15}$$

$$\cos(\theta) = 1 - \frac{d^2}{2} \tag{16}$$

The signed cosine similarity still exhibits a quadratic relationship with the Euclidean distance. The considerations on the convergence of the cosine similarity remain valid.

## A.3 EFFECT OF VECTORS' DISTRIBUTION

The effect of dimensionality on randomly generated vectors $(\mathbf{a}, \mathbf{b})$ was tested on elements drawn from three different distributions: (i) uniform (Figure 2.a), (ii) Gaussian (Figure 9.a), (iii) uniformly distributed on a unit sphere (Figure 9.b) (Marsaglia, 1972; Muller, 1959).

The same algorithm proposed in Figure 1 was adopted. In the Gaussian distribution case, the elements of the vectors $(\mathbf{a}, \mathbf{b})$ were sampled from Gaussian distributions with mean and standard deviation respectively equal to: Real positive: $\mu = 0.5, \sigma = 0.3$, Real negative $\mu = -0.5, \sigma = 0.3$, All real $\mu = 0, \sigma = 0.6$.

In the case of a uniform unit sphere distribution, the elements of the vectors $(\mathbf{a}, \mathbf{b})$ were sampled from a uniform distribution following the approach presented in the main manuscript (Section *Effects of Vectors' Dimensionality*) and then projected onto a unit $n$-sphere using the algorithm proposed in the works of Marsaglia and Muller (Marsaglia, 1972; Muller, 1959).

Figure 9 aligns with the results presented in Figure 2.a, confirming that the cosine similarity metric shows a converging trend with decreasing variance for every tested distribution. The reason for this behavior can be attributed to the normalization of the vectors $(\mathbf{a}, \mathbf{b})$ in the mathematical formulation

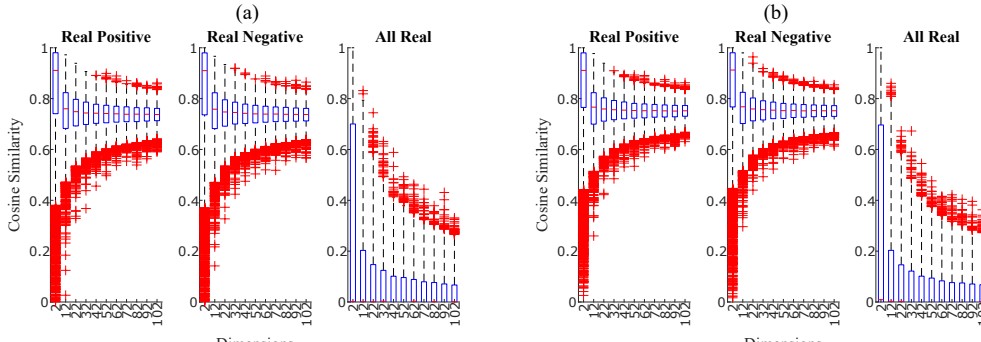

Figure 9: Panel (a): Cosine similarity boxplots for increasing dimension of the vectors **a** and **b**. The elements of the vectors were sampled from a Gaussian distribution. Panel (b):Cosine similarity boxplots for increasing dimension of the vectors **a** and **b**. The elements of the vectors were sampled from a uniform unit n-sphere distribution. The three sub-panels show, respectively, the case in which vectors elements were only positive (left), only negative (center) or could assume all real values within the given range (right).

of cosine similarity (Equation 1). This normalization causes the considered distributions to "collapse" onto a unit $n$-dimensional sphere.

In the authors' opinion, a uniform distribution is the most representative of a real scenario, especially when the vectors represent physical quantities. For example, if the elements of the vectors **a**, **b** are physical quantities—such as force or position—acquired through sensors, they will arguably have the same probability of being sampled between the minimum $v_m$ and maximum $v_M$ of the sensor sampling range, thus justifying the uniform distribution. However, for completeness, we explored distributions that might occur in different data processing conditions.

Figure 10 provides an intuitive bi-dimensional understanding of the differences between vectors (or points) sampled from the uniform, Gaussian, and uniformly distributed unit sphere distributions. Despite these different distributions, Figure 9 and Figure 2 show that the main trends used to define DIEM were observed in all cases.

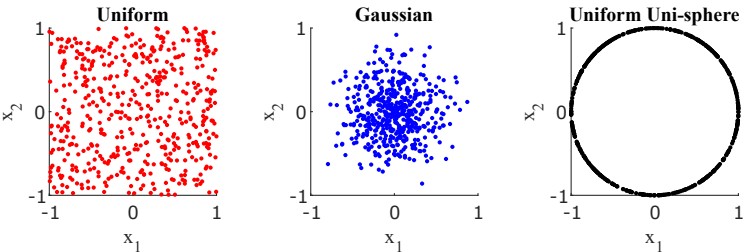

Figure 10: Two dimensional points sampled with from three different distributions. The left plot (red dots) shows points sampled from a uniform distribution $U(-\mathbf{1}, \mathbf{1})$. The central plot (blue dots) shows points sampled from a Gaussian distribution with mean 0 and standard deviation 0.3. The right plot (black dots) shows points sampled from a uniform distribution on the unit sphere ($r = 1$).

A.4    TRANSITION FROM NON-NORMAL TO NORMAL DISTRIBUTION

We observed that the distribution of Euclidean distances between randomly generated vectors tended to become normal with the increase in the vectors' dimension. Here, we present the results of a finer simulation that shows the transition between non-normal and normal distributions. All three cases (Real Positive, Real Negative, and All Real) were considered and yielded equivalent results.

The simulation was performed considering vector dimensions spanning from $n = 2$ to $n = 12$. Kolmogorov-Smirnov tests were conducted to check whether the observed simulated distribution was

significantly different from a normal distribution with mean and standard deviation equal to those of the simulated data, i.e., $\mathcal{N}(\text{mean}(d), \text{std}(d))$. A $p$-value of $p < 0.05$ was consistently observed for distributions with $n < 5$, while it was consistently $p > 0.05$ for $n > 7$.

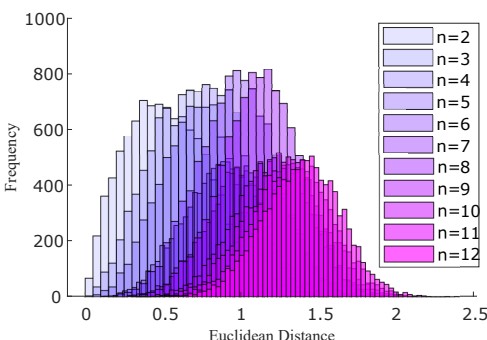

Figure 11: Histograms of the non-normalized Euclidean distance for growing dimensions n in the range $2 \leq n \leq 12$.

Figure 11 presents the trend of the Euclidean distance distribution for the Real Positive case in the range $2 \leq n \leq 12$. The results confirm a smooth transition from a non-normal to a normal distribution for $n > 7$.

### A.5 EFFECT OF DIMENSIONS ON MANHATTAN DISTANCE

Considering two vectors $\mathbf{a}$, $\mathbf{b}$, each composed of $n$ elements, the Manhattan distance is defined as:

$$d_M = \sum_{i=1}^{n} |a_i - b_i| \tag{17}$$

Applying the same algorithm presented in Figure 1, Figure 12 presents the Manhattan distance evolution with increasing dimensions ($n$). The Manhattan distance grows linearly with respect to the number of dimensions, and this can be demonstrated by calculating the maximum Manhattan distance value between two random vectors and realizing that it grows proportionally to $n$:

$$d_{M\max}(n) = n\,|v_{\max} - v_{\min}| \tag{18}$$

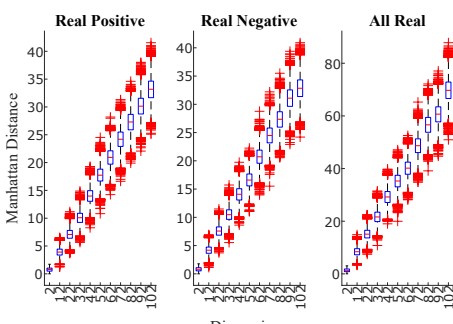

Figure 12: Manhattan distance for increasing dimension of the vectors $\mathbf{a}$ and $\mathbf{b}$. The three panels show, respectively, the case in which vectors elements were only positive (left), only negative (center) or could assume all real values within the given range (right).

More importantly, the variance of the Manhattan distance does not remain constant with the increase in dimensions as it does for the Euclidean distance. It actually grows with dimensions. This is a

limitation of its applicability for a dimension-insensitive metric compared to the Euclidean metric presented in this work. Higher-order norms, e.g., 3-norm, 4-norm, and so on, could be attempted, but that investigation is not treated here.

### A.6 PROOF OF CONVERGENCE OF COSINE SIMILARITY

#### A.6.1 ALL REAL CASE

Consider two vectors $\mathbf{a}, \mathbf{b} \in \mathbb{R}^n$ and assume these vectors are independently sampled on the uniform unit-sphere such that $\|\mathbf{a}\| = \|\mathbf{b}\| = 1$.

The cosine similarity, i.e., the angle between these vectors, follows Equation 1 (unsigned version) or Equation 14 (signed version). For simplicity, but without loss of accuracy, we consider the form in Equation 14:

$$\cos\left(\theta\right) = \frac{\mathbf{a}^T \cdot \mathbf{b}}{\|\mathbf{a}\| \cdot \|\mathbf{b}\|} = \mathbf{a}^T \cdot \mathbf{b} = \sum_{i=1}^{n} a_i b_i \tag{19}$$

Since $\mathbf{a}$ and $\mathbf{b}$ are i.i.d. uniformly distributed on the unit sphere centered around 0, each of their entries $a_i$ and $b_i$ has zero mean:

$$E[a_i] = E[b_i] = 0 \tag{20}$$

This is due to the symmetry of the unit sphere. As a consequence, the expectation of the inner product, i.e., the cosine of their angle, is equal to:

$$E[\mathbf{a}^T \cdot \mathbf{b}] = E\left[\sum_{i=1}^{n} a_i b_i\right] = \sum_{i=1}^{n} E[a_i]E[b_i] = 0 \tag{21}$$

This is already sufficient to show that the expected value of the cosine similarity between uniformly distributed points on a unit-sphere is 0, and thus their resulting angle is 90 degrees.

To show the convergence for higher dimensions, we compute the variance of the cosine similarity:

$$\text{Var}(\mathbf{a}^T \cdot \mathbf{b}) = E\left[\left(\mathbf{a}^T \cdot \mathbf{b}\right)^2\right] - \left(E[\mathbf{a}^T \cdot \mathbf{b}]\right)^2 = E\left[\left(\mathbf{a}^T \cdot \mathbf{b}\right)^2\right] \tag{22}$$

Expanding the square of the inner product, we obtain:

$$(\mathbf{a}^T \cdot \mathbf{b})^2 = \left(\sum_{i=1}^{n} a_i b_i\right)^2 = \sum_{i=1}^{n} a_i^2 b_i^2 + 2\sum_{i<j}^{n} a_i b_i a_j b_j \tag{23}$$

Taking expectations:

$$\begin{aligned} E\left[\left(\mathbf{a}^T \cdot \mathbf{b}\right)^2\right] = E\left[\left(\sum_{i=1}^{n} a_i b_i\right)^2\right] &= E\left[\sum_{i=1}^{n} a_i^2 b_i^2 + 2\sum_{i<j}^{n} a_i b_i a_j b_j\right] \\ &= E\left[\sum_{i=1}^{n} a_i^2 b_i^2\right] + 2E\left[\sum_{i<j}^{n} a_i b_i a_j b_j\right] \end{aligned} \tag{24}$$

Since $a_i$ and $b_i$ are i.i.d., we have:

$$E\left[\sum_{i<j}^{n} a_i b_i a_j b_j\right] = 0 \tag{25}$$

Hence:

$$E\left[\left(\mathbf{a}^T \cdot \mathbf{b}\right)^2\right] = E\left[\sum_{i=1}^{n} a_i^2 b_i^2\right] = \sum_{i=1}^{n} E[a_i^2]E[b_i^2] \tag{26}$$

However, since $\mathbf{a}$ and $\mathbf{b}$ lie on the unit sphere, their norms satisfy:

$$\|\mathbf{a}\|^2 = \|\mathbf{b}\|^2 = \sum_{i=1}^{n} a_i^2 = \sum_{i=1}^{n} b_i^2 = 1 \tag{27}$$

Therefore:

$$E\left[\sum_{i=1}^{n} a_i^2\right] = nE[a_i^2] = 1 \implies E[a_i^2] = \frac{1}{n} = E[b_i^2] \tag{28}$$

Substituting Equation 27 into Equation 23, we obtain:

$$E\left[\left(\mathbf{a}^T \cdot \mathbf{b}\right)^2\right] = n\left(\frac{1}{n} \cdot \frac{1}{n}\right) = \frac{1}{n} \tag{29}$$

We finally obtain that the variance of the cosine similarity between $\mathbf{a}$ and $\mathbf{b}$ is equal to:

$$Var\left(\mathbf{a}^T \cdot \mathbf{b}\right) = \frac{1}{n} \tag{30}$$

According to Equation 29, the variance of the cosine similarity for uniformly sampled vectors on the unit-sphere decreases with the increase of dimensions. Moreover, from the Central Limit Theorem, we can write that the cosine similarity assumes a normal distribution with mean 0 and variance $1/n$:

$$\cos(\theta) = \mathbf{a}^T \cdot \mathbf{b} \sim N\left(0, \frac{1}{n}\right) \tag{31}$$

For completeness, we demonstrate the convergence of this distribution for $n \to \infty$. To do this, we can consider Chebyshev's inequality for a random variable $X$ with mean $\mu$ and variance $\sigma^2$:

$$\Pr\left(|X - \mu| \geq k\sigma\right) \leq \frac{1}{k^2}, \ \forall k \in \mathbb{R}^+ \tag{32}$$

Applying Equation 31 to our cosine similarity distribution, we obtain:

$$\Pr\left(|\cos(\theta)| \geq \alpha\right) = \Pr\left(|\mathbf{a}^T \cdot \mathbf{b}| \geq \alpha\right) \leq \frac{\sigma^2}{\alpha^2}, \quad \text{with } \alpha = k\sigma$$
$$\implies \Pr\left(|\cos(\theta)| \geq \alpha\right) \leq \frac{Var\left(\mathbf{a}^T \cdot \mathbf{b}\right)}{\alpha^2} = \frac{1}{n\alpha^2} \tag{33}$$

The limit for $n \to \infty$ of $\frac{1}{n\alpha^2}$ is equal to 0, while $\Pr(\cdot) \geq 0$. As a consequence:

$$\lim_{n\to\infty} \Pr\left(|\cos(\theta)| \geq \alpha\right) = 0, \quad \forall \alpha \in \mathbb{R}^+ \implies \lim_{n\to\infty} |\cos(\theta)| = 0 \tag{34}$$

Therefore, as $n \to \infty$, the vectors $\mathbf{a}$ and $\mathbf{b}$ will be orthogonal with shrinking variance. This demonstrates the convergence observed in Figure 2.a for the all-real case.

### A.6.2 REAL POSITIVE CASE

Unlike the all-real case, we now consider the elements of the vectors $\mathbf{a}$ and $\mathbf{b}$ sampled as i.i.d. of the type $a_i, b_i \sim U(0, 1)$. In this case, the vectors are no longer unit vectors, and the cosine similarity remains equal to Equation 14: $\cos(\theta) = \frac{\mathbf{a}^T \cdot \mathbf{b}}{||\mathbf{a}|| \cdot ||\mathbf{b}||}$

Recalling the generic uniform distribution $X \sim U(a, b)$, we know that:

$$E[X] = \frac{1}{2}(a+b), \quad Var(X) = \frac{1}{12}(b-a)^2 \tag{35}$$

Therefore, applying Equation 34 to our case, we obtain:

$$E[a_i] = E[b_i] = \frac{1}{2}, \quad Var(a_i) = Var(b_i) = \frac{1}{12} \tag{36}$$

We can now compute the expectation as follows:

$$
\begin{aligned}
E[a_i b_i] &= E[a_i] E[b_i] = \frac{1}{4}, \\
E[a_i^2] &= E[b_i^2] = \int_0^1 a^2 da = \frac{1}{3}, \\
E[(a_i b_i)^2] &= E[a_i^2] E[b_i^2] = \frac{1}{9}, \\
E[a_i^3] &= E[b_i^3] = \int_0^1 a^3 da = \frac{1}{4}, \\
E[a_i^4] &= E[b_i^4] = \int_0^1 a^4 da = \frac{1}{5}
\end{aligned}
\tag{37}
$$

The variances will then result in:

$$
\begin{aligned}
Var(a_i b_i) &= E[(a_i b_i)^2] - (E[a_i b_i])^2 = \frac{1}{9} - \left(\frac{1}{4}\right)^2 = \frac{7}{144}, \\
Var(a_i^2) &= Var(b_i^2) = E[(a_i^2)^2] - (E[a_i^2])^2 = \frac{1}{5} - \frac{1}{9} = \frac{4}{45}
\end{aligned}
\tag{38}
$$

To simplify the cosine similarity expression, we define the following quantities:

$$A_n = \sum_{i=1}^n a_i b_i, \quad B_n = \sum_{i=1}^n a_i^2, \quad C_n = \sum_{i=1}^n b_i^2 \tag{39}$$

Then, the cosine similarity equation becomes:

$$\cos(\theta) = \frac{\mathbf{a}^T \cdot \mathbf{b}}{||\mathbf{a}|| \cdot ||\mathbf{b}||} = \frac{A_n}{\sqrt{B_n C_n}} \tag{40}$$

We can now decompose each term in Equation 39 into its expected value plus a fluctuation term to analyze the behavior for $n \to \infty$:

$$
\begin{aligned}
A_n &= nE[a_i b_i] + \delta_{A_n} = \frac{n}{4} + \delta_{A_n}, \\
B_n &= nE[a_i^2] + \delta_{B_n} = \frac{n}{3} + \delta_{B_n}, \\
C_n &= nE[b_i^2] + \delta_{C_n} = \frac{n}{3} + \delta_{C_n}
\end{aligned}
\tag{41}
$$

The associated fluctuation terms are:

$$
\delta_{A_n} = \sum_{i=1}^{n} \left( a_i b_i - E\left[a_i b_i\right] \right),
$$

$$
\delta_{B_n} = \sum_{i=1}^{n} \left( a_i^2 - E\left[a_i^2\right] \right), \tag{42}
$$

$$
\delta_{C_n} = \sum_{i=1}^{n} \left( b_i^2 - E\left[b_i^2\right] \right)
$$

The fluctuations $\delta_{A_n}$, $\delta_{B_n}$, and $\delta_{C_n}$ are sums of i.i.d. random variables with zero mean. Therefore, according to the Central Limit Theorem, the sum of a large number of i.i.d. random variables with finite mean and variance will be approximately normally distributed, regardless of the original distribution of the variables. Mathematically, for i.i.d. random variable $z_i$ with mean $\mu$ and standard deviation $\sigma$, its sum $S = \sum_{i=1}^{n} z_i$ satisfies:

$$
\lim_{n \to \infty} \frac{S - n\mu}{\sqrt{n}\sigma} \sim N(0, 1) \tag{43}
$$

Using the sum of i.i.d. random variables, $Var(X + Y) = Var(X) + Var(Y)$, we calculate the standard deviations of the terms $A_n$, $B_n$, and $C_n$ as follows:

$$
\sigma_A = \sqrt{nVar\left(a_i b_i\right)} = \frac{\sqrt{7n}}{12},
$$

$$
\sigma_B = \sigma_C = \sqrt{nVar\left(a_i^2\right)} = 2\sqrt{\frac{n}{45}} \tag{44}
$$

Since we only care about the order of the fluctuations, we can consider that:

$$
\delta_{A_n} = \delta_{B_n} = \delta_{C_n} = O\left(\sqrt{n}\right) \tag{45}
$$

Hence:

$$
\frac{\delta_{A_n}}{n} = \frac{\delta_{B_n}}{n} = \frac{\delta_{C_n}}{n} = O\left(\frac{1}{\sqrt{n}}\right) \tag{46}
$$

We can now expand the cosine similarity presented in Equation 39 using Equation 40:

$$
\cos(\theta) = \frac{A_n}{\sqrt{B_n C_n}} = \frac{\frac{n}{4} + \delta_{A_n}}{\sqrt{\left(\frac{n}{3} + \delta_{B_n}\right)\left(\frac{n}{3} + \delta_{C_n}\right)}} \tag{47}
$$

The denominator of Equation 47 can be further simplified by re-expressing it in a different form and using the binomial approximation expansion truncated at the first-order terms $\left(\sqrt{(1+b)(1+c)} \approx 1 + \frac{b+c}{2}\right)$:

$$
\sqrt{B_n C_n} = \frac{n}{3}\sqrt{\left(1 + \frac{3\delta_{B_n}}{n}\right)\left(1 + \frac{3\delta_{C_n}}{n}\right)} \approx \frac{n}{3}\left(1 + \frac{3}{2n}\left(\delta_{B_n} + \delta_{C_n}\right)\right) = \frac{n}{3} + \frac{1}{2}\left(\delta_{B_n} + \delta_{C_n}\right) \tag{48}
$$

Plugging Equation 48 into Equation 47 we obtain:

$$\cos\left(\theta\right) = \frac{A_n}{\sqrt{B_n C_n}} \approx \frac{\frac{n}{4} + \delta_{A_n}}{\frac{n}{3} + \frac{1}{2}\left(\delta_{B_n} + \delta_{C_n}\right)} = \frac{\frac{1}{4} + \frac{\delta_{A_n}}{n}}{\frac{1}{3} + \frac{1}{2}\left(\frac{\delta_{B_n}}{n} + \frac{\delta_{C_n}}{n}\right)} \tag{49}$$

From Equation 49, we can immediately see that the cosine similarity converges to $\frac{3}{4}$ as $n$ approaches $\infty$. For a more rigorous analysis of the convergence rate, we can now consider the variance of the cosine similarity.

For clarity, we define:

$$\alpha = \frac{\delta_{A_n}}{n}, \quad \beta = \frac{\delta_{B_n}}{n}, \quad \gamma = \frac{\delta_{C_n}}{n} \tag{50}$$

Therefore, Equation 49 becomes:

$$\cos\left(\theta\right) \approx \frac{\frac{1}{4} + \alpha}{\frac{1}{3} + \frac{1}{2}(\beta + \gamma)} \tag{51}$$

The denominator of Equation 51 can be further simplified by using the Taylor series expansion truncated at the first-order terms:

$$\frac{1}{3} + \frac{1}{2}(\beta + \gamma) = \frac{1}{3}\left(1 + \epsilon\right), \quad \text{with } \epsilon = \frac{3}{2}(\beta + \gamma) \tag{52}$$

Applying the first-order approximation:

$$\frac{1}{1 + \epsilon} \approx 1 - \epsilon = 1 - \frac{3}{2}(\beta + \gamma) \tag{53}$$

Thus,

$$\frac{1}{\frac{1}{3}(1 + \epsilon)} \approx 3 - \frac{9}{2}(\beta + \gamma) \tag{54}$$

We can now substitute Equation 54 into Equation 51 to obtain:

$$\cos\left(\theta\right) \approx \left(\frac{1}{4} + \alpha\right)\left(3 - \frac{9}{2}(\beta + \gamma)\right) \approx \frac{3}{4} + 3\alpha - \frac{9}{8}(\beta + \gamma) \tag{55}$$

In Equation 55, we neglected the higher-order terms of the binomial product as they will diminish faster than first-order terms. Remember that the orders of $\alpha$, $\beta$, and $\gamma$ are $O\left(\frac{1}{\sqrt{n}}\right)$ (refer to Equation 46). Therefore, the order of the standard deviation of cosine similarity (Equation 55) will also be $O\left(\frac{1}{\sqrt{n}}\right)$. This indicates that for $n \to \infty$, $\cos\left(\theta\right)$ converges to $\frac{3}{4} = 0.75$, which is the same result observed in Figure 2.a for the real positive case.

The real negative case does not need to be proven, as the reasoning will follow the same logic as presented above.