# OpenReview forum: "Surpassing Cosine Similarity  for Multidimensional Comparisons:  Dimension Insensitive Euclidean Metric"
_ICLR.cc/2026/Conference — ICLR 2026 Conference Withdrawn Submission_

### Official Review · Reviewer_9Rbz · 2025-10-18

**Soundness:** 2
**Presentation:** 3
**Contribution:** 1
**Rating:** 2
**Confidence:** 4

**Summary:**

The paper proposes DIEM, a Dimension Insensitive Euclidean Metric. DIEM is a metric based on the Euclidean distance, where the expected value is subtracted and scaled by a factor based on the data's standard deviation and range.

**Strengths:**

Section 2 is well written and precise.

The values for DIEM_{min} and DIEM_{max} are theoretically derived.

**Weaknesses:**

Avoid the word "replace"; you are adding a new alternative. Cosine similarity is a well-established metric, with several important characteristics.

Even though the general problem is clearly described, there is no explanation of the real issues with the cosine distance. This could be added in the introduction, rather than going to the third section to explain the problem.

The related work must be updated. New metrics and characteristics are appearing periodically. Unfortunately, the newest reference is from 2023. Also, please clean the references. I do not believe that a prize should be in the reference. Also, several references contain repeated information (especially the year), incorrect links (a DOI with a JPG extension), and other issues.

Figure 1 must be explained in the text. Figure 1 does not describe the main idea of the simulation. Instead, the paper forces the reader to understand the simulation based exclusively on the figure.

Equation 7 assumes that all the variables have the same range. The paper can change the = to <= for better generalization.

Instead of leaving the work to the readers, the paper can use the appendix to demonstrate the step from equation 8 to equation 9.

The curse of dimensionality is well known for all types of metrics. Several results are already known in the literature (lack of novelty).

There is an important theoretical issue in the paper that makes its publication impossible. DIEM is proposed as a metric, and a metric has four properties, where three of them seem to be violated: non-negativity (violated by figure 5), symmetry (can be deduced that is violated by figure 5), and triangle inequality (can be deduced that is violated by figure 5). Also, if DIEM is a similarity/dissimilarity measure, the value must be positive, but this is not the case, as shown in Figure 5.

The explanation of the proposed must be improved. Equation 11 refers to a constant value, as DIEM is not based on vectors a and b (DIEM(a,b)). \sigma^2_{ed} is barely defined as the variance of the Euclidean distance. There is no formula or empirical explanation for the calculation of E[d(n)] for a given dataset.

The paper changes the focus in Figure 6 to a similarity/dissimilarity measure.

The paper should also consider whether the proposed metric addresses one of the many issues of the curse of dimensionality, where all the data points are at the same distance as the dimensionality increases. Given the formula, which mainly involves changes in the mean and the standardization of the values, this metric does not seem to solve this issue.

The results from Figure 7 show a clear disadvantage of the proposed metric, and further analysis must be applied. First, it calls my attention the big difference between the Pairwise comparison (Cosine vs DIEM). It will be interesting to analyze cases where Cosine similarity equals zero and DIEM has a significant negative value. Second, if everything is compared and the dataset shows differences between the phrases (for that reason, only a couple of phrases are paired), there is a high probability that the phrases will not be related, as depicted in Figure 7 for Cosine similarity. In contrast, DIEM shows, with high probability, that they are quite similar. A further evaluation of these results is needed to explain these results.

Also, as a suggestion, the paper should include other metrics, such as the Mahalanobis distance, which also accounts for the variability of the data points.

**Questions:**

How does the paper calculate \sigma^2_{ed} in a real case scenario, where you have a specific number of data points?

How does the paper calculate E[d(n)] in a real case scenario, where you have a specific number of data points?
In line 321, what does it mean "This facilitate precise statistical testing"? What type of statistical testing? Please, the paper must elaborate on this idea.

Given a dataset, does the method have to calculate all the Euclidean distances to estimate a single metric? If this is the case, how does the number of data points affect this metric?

---

### Official Review · Reviewer_9ZTP · 2025-10-29

**Soundness:** 2
**Presentation:** 3
**Contribution:** 1
**Rating:** 0
**Confidence:** 5

**Summary:**

The paper highlights the effect of dimensionality on the cosine similarity and on the Euclidean distance. Specifically, as the dimensionality grows, both measures begin to concentrate. The authors then propose that, to make the interpretation of high-dimensional computations more intuitive, one can subtract the expected value of the Euclidean distance. The argument is that this "de-trended" version of the Euclidean metric then behaves consistently with one's low-dimensional intuition.

**Strengths:**

I really like this paper's stated objective. It is certainly the case that metrics adopt unexpected, often bizarre properties in high-dimensional spaces. It is also the case that these properties are poorly understood among the general population and that having sound explanations of them would be extremely useful. The present paper satisfies both of these criteria and does so effectively -- it is well-written, clearly stated and the examples naturally build on each other. In these contexts, I think the paper does a great job.

**Weaknesses:**

Unfortunately, I do not believe this work is sufficiently novel as to be a full contribution on its own. Although the description of high-dimensional similarity behaviors is well-written, it is not novel. There are blog posts dedicated to this subject (such as [1], [2], for instance). Indeed, the term "curse of dimensionality" is famous specifically for describing this phenomenon. Thus, the analysis of high-dimensional distributions (which takes up several pages of space in the paper) is working through already-known effects.

Similarly, the idea that one should "correct" these issues has been analyzed. However, I think the community has largely accepted that it is *not necessary* to correct these issues. Actually, it seems that these properties of high-dimensional spaces are extraordinarily *useful*. For example, if the cosine similarity between two vectors is so strongly concentrated around 0, then it means that these vectors are largely orthogonal. Indeed, the famed johnson-lindenstrauss lemma exploits precisely this property! Without this property, the JL lemma and all its use-cases would not exist. On top of this, the entire field of superposition in mechanistic interpretability suggests that this mutual orthogonality drives the representational capacity of LLMs. Similarly, it turns out the cosine similarity has many interesting properties when optimized via gradient descent (for example, as shown in [3]).


In short, while I commend the paper's intentions, I do not believe that it is sufficiently novel or has a sufficient contribution to warrant acceptance. I really like the work -- it would make an *excellent* blog post. But to make this a paper at the ICLR level, it would require showing a clear utility that was previously absent from the literature. I hope this constructive criticism is useful.


[1]: https://www.inference.vc/high-dimensional-gaussian-distributions-are-soap-bubble/

[2]: https://medium.com/@prathik.codes/why-does-euclidean-distance-fail-in-high-dimensions-a0a0748784a3

[3]: https://icml.cc/virtual/2025/poster/45610

**Questions:**

Towards restructuring this paper to make the contribution more novel/relevant towards downstream use-cases, how would you change the discussion around the DIEM to show that it necessarily improves the interpretability of high-dimensional vectors? Is there something that one can do better with this new metric? Does it modify the relationships between points in an exploitable way?

---

### Official Review · Reviewer_gzDt · 2025-11-02

**Soundness:** 3
**Presentation:** 3
**Contribution:** 2
**Rating:** 4
**Confidence:** 4

**Summary:**

This paper proposes DIEM, Dimension-Insensitive Euclidean Metric as expressed by equation (11), as a replacement of cosine similarity. Cosine similarity has several drawbacks especially for high dimensions; especially, its practical range will be different for different dimensions. Contrary to cosine similarity, DIEM has always the same practical ranges (but not within [-1,1]), and experimental results for LLM embeddings reveal that DIEM clearly behaves better than cosines when we also consider random embeddings. Empirical results are limited to only this case, thus more practical experiments could be desired.

**Strengths:**

It is widely known that cosine similarity does not have full support over [-1,1] in practice. The proposed DEIM metric removes this limitation and becomes a more suitable distance for measuring embeddings. Experimental results including random embeddings are interesting and important, and differences shown in Figure 7 between all combinations over sentence embeddings and random embeddings are also worth noting.

**Weaknesses:**

While the proposed DEIM is interesting and important, it seems that this paper still needs more experiments and theoretical considerations.

- Effectiveness of DEIM is only shown for LLM: this will not be sufficient, because cosine similarities are integrated on many tasks in natural language processing and information retrieval. I would like to see more results on various tasks on NLP, especially for information retrieval results (such as precision-recall on sentence embeddings, not only the distributions shown in Figure 7).

- As a whole, I am concerned that there are very little arguments with respect to high-dimensional statistics. It is exactly the theme of this kind of phenomenon in high-dimensional statistics, and the theories and results there could also be beneficial for building DEIM and showing its theoreical properties.

- DEIM expression needs an expectation E[d(n)] in equation (11), but equation (8) and (9) only show an upper bould of this value. Is this practically justified? How much difference there are when we just replace the expectation with the upper bound or numerical simulation?

Finally, restricting cosine similarity to nonegative values by taking absolute values is not common in NLP. If you wish to stick to positiveness, more persuasive arguments would be necessary. Also, I would like to see the proposed similarity in the standard scale: what is the meaning of "-120" in Figure 7? Multiplying by the input scale v_M - v_m in equation (11) could be removed or replaced by another expression to yield a standardized scale.

**Questions:**

See above.

---

### Official Review · Reviewer_e5Tq · 2025-11-03

**Soundness:** 2
**Presentation:** 2
**Contribution:** 2
**Rating:** 2
**Confidence:** 4

**Summary:**

This paper analyzes the behavior of cosine similarity in high-dimensional spaces, showing that it converges to fixed values as dimensionality increases (eg 0.75 if all $x_i$s are positive/negatives, though the authors need to show which real world embedding models or datasets this assumption applies to). The authors propose DIEM (Dimension Insensitive Euclidean Metric) as an alternative that maintains consistent variance across dimensions by detrending Euclidean distance. They validate their approach through simulations and an experiment with LLM text embeddings.

**Strengths:**

* The paper provides thorough theoretical proofs of cosine similarity convergence and derives analytical bounds for expected Euclidean distances. The mathematical framework appears well-developed.
* The visualization of dimension-dependent behavior (Figure 2 and 5) effectively illustrates the convergence issue and how DIEM addresses it.
* The authors provide code and use publicly available datasets, supporting reproducibility.

**Weaknesses:**

* The paper's core premise is problematic. The authors present it as a problem that unrelated text embeddings have similar cosine similarity to random vectors and use it to justify DIEM's superiority in Section 6, but AFAIK this is actually correct behavior. In a well-functioning embedding space, semantically unrelated content should be as dissimilar as random noise. The paper fails to justify why distinguishing between "unrelated" and "random" matters practically.
* The paper provides no evidence that DIEM improves performance on actual tasks (classification, clustering, etc.). The LLM experiment shows statistical differences but doesn't demonstrate that these differences translate to better outcomes. It's also over a single embedding model.
* Convergence of cosine similarity in high dimensions is well-known (concentration of measure phenomenon). The paper doesn't engage with existing literature on why cosine similarity remains effective despite this property, nor does it explain why many high-dimensional applications (modern NLP, deep learning) succeed with cosine similarity.
* Overall, the assertion that DIEM "has the potential to replace cosine similarity" is not sufficiently supported by the evidence presented.

**Questions:**

* Can you provide concrete real-world use cases where distinguishing between "unrelated content" and "random noise" provides practical value?
* How does DIEM perform on standard benchmarks, eg information retrieval or clustering tasks?
* Are there specific domains where dimension-sensitive variance matters?

---

### Note · Authors · 2025-11-21

**Comment:**

Dear Editorial office of ICLR,
Dear Reviews,

After careful evaluation of the reviewers' comments, we decided to withdraw our manuscript.

We believe the reviewers' comments were informative, constructive, and helpful.

However, we did not have enough time to address the comments properly, and therefore we decided to not proceed with further evaluation of our paper.

Nonetheless, we will definitely implement the reviewers' comments and we thank them for their time to read our paper and provide their comments.

Kind regards,

The authors

**Withdrawal Confirmation:**

I have read and agree with the venue's withdrawal policy on behalf of myself and my co-authors.